# OpenReview forum: "Robust LLM Unlearning via Post Judgment and Multi-round Thinking"
_ICLR.cc/2026/Conference — ICLR 2026 Poster_

### Official Review · Reviewer_cWjy · 2025-10-24

**Soundness:** 3
**Presentation:** 3
**Contribution:** 3
**Rating:** 6
**Confidence:** 3

**Summary:**

The paper highlights that existing pre-filtering unlearning methods fail under adversarial attacks, causing severe leakage and utility loss. To address this, the authors propose PoRT, a framework combining dynamic data cleaning, post-judgment evaluation, and multi-round self-correction. Experiments show PoRT is more robust against attacks and has strong unlearning effectiveness while preserving model utility.

**Strengths:**

- PoRT introduces a post-judgment mechanism that jointly analyzes prompts and responses, effectively leveraging LLMs’ reasoning capabilities for more reliable unlearning.

- The paper provides extensive empirical evaluation, demonstrating robustness against adversarial attacks and effective unlearning on TOFU and WMDP, with superior performance compared to SOTA methods.

- The overall writing is clear and well-structured.

**Weaknesses:**

- It would be helpful to elaborate on the comparison with model-based unlearning methods (Line 39), particularly regarding effectiveness and efficiency against adversarial attacks and on unlearning tasks.

- Providing statistical uncertainties for the reported results would strengthen the empirical evaluation and make the findings more convincing.

- Minor comment: The definitions of Forget Probability, ROUGE, and 1-TR for Figure 2 appear only in the appendix (Line 310). A brief, early description would help readers follow the results without jumping ahead.

**Questions:**

Please see **Weaknesses** for my questions.

---

> ### Author Response · Authors · 2025-11-20
>
> Dear Reviewer cWjy,
>
> We thank you for your insightful feedback and positive assessment. We are grateful for your recognition of our work's **novelty, empirical rigor, and clear presentation**. We address each of your valuable suggestions below.
>
> > **W1:** It would be helpful to elaborate on the comparison with model-based unlearning methods (Line 39), particularly regarding effectiveness and efficiency against adversarial attacks and on unlearning tasks.
>
> Comprehensive comparative results with model-based methods **have been detailed in the Appendix D.4 & D.5 of our original submission** (e.g., Tables 7, 9, and 10). To facilitate your review, we summarize the key findings again below, focusing on the TOFU 5% split and WMDP-Bio subset under Noise Prefix attacks:
>
>
> * **Performance on TOFU (5% Split, on Llama-2-7b-chat-hf):** Under standard  conditions, PoRT outperforms all model-based baselines. This performance gap widens under attacks.
>
>     | Method        | MU (Standard) $\uparrow$ | HFQ (Standard) $\uparrow$ | HFQ (Under Attack) $\uparrow$ |
>     | :-------------- | :---: | :---: | :---: |
>     | Original      | 66.58% | 14.07% | 9.48% (-4.59%) |
>     | Retain        | 67.35% | 99.34% | 97.83% (-1.51%) |
>     | GA   | 22.19% | 9.43%  | 9.02% (-0.41%)  |
>     | GD    | 55.83% | 16.20% | 13.77% (-2.43%) |
>     | RMU           | 60.00% | 50.95% | 33.94% (-17.01%)|
>     | NPO           | 50.35% | 56.54% | 31.25% (-25.29%)|
>     | SimNPO        | 65.33% | 48.80% | 39.03% (-9.77%) |
>     | **Ours (PoRT)** | **67.21%** | **84.74%** | **84.41% (-0.33%)** |
>
> * **Performance on WMDP (Bio Subset, on Zephyr-7b-beta):** The WMDP benchmark further highlights PoRT's unique strengths. We compare the forget effectiveness (WMDP-Bio Accuracy) and model utility (MMLU score).
>
>     | Method        | MMLU (Standard) $\uparrow$ | WMDP-Bio (Standard) † | WMDP-Bio (Under Attack) † |
>     | :-------------- | :---: | :---: | :---: |
>     | Original      | 58.9% | 64.3% | 63.7% |
>     | GA            | 57.3% | 62.0% | 62.4% |
>     | GD            | 52.9% | 56.6% | 57.2% |
>     | NPO           | 57.5% | 62.0% | 62.3% |
>     | SimNPO        | 49.0% | 46.5% | 48.0% |
>     | RMU           | 57.5% | 29.5% | 37.4% |
>     | **Ours (PoRT)** | **58.9%** | **25.1%** | **24.1%** |
>
>     †: For WMDP subsets, best results (closest to 25%) are bolded. $\uparrow$: For MMLU, higher results are bolded.
>
> Due to space constraints, the full results were placed in the appendix of our original submission, with key conclusions in the main text. We have elevated the prominence of these key comparisons in Section 4.2 & 4.3 of the revised manuscript for improved clarity.
>
> ***

---

> ### Author Response · Authors · 2025-11-20
>
> > **W2:** Providing statistical uncertainties for the reported results would strengthen the empirical evaluation and make the findings more convincing.
>
> We thank you for this excellent suggestion. Our initial experiments used a fixed seed to align with common practice in prior unlearning papers[1, 2, 3, 4] and ensure strict reproducibility. To further demonstrate the reliability of our results and provide statistical uncertainties, we also report a multi-seed analysis over 5 different random seeds below.
>
> For ease of review, we present the most essential results for both the TOFU and WMDP benchmarks below, with key metrics updated to **mean ± standard deviation**.
>
> * **Results on TOFU (5% Split, Llama-2-7b-chat-hf)**
>
>     | Method | HFQ $\uparrow$ | MU $\uparrow$ |
>     | :--- | :---: | :---: |
>     | Original | $0.1415_{±0.0132}$ | $0.6662_{±0.0054}$ |
>     | Retain | $0.9928_{±0.0087}$ | $0.6731_{±0.0021}$ |
>     | Grad Ascent | $0.1054_{±0.0821}$ | $0.2187_{±0.0314}$ |
>     | Grad Diff | $0.1635_{±0.0045}$ | $0.5596_{±0.0192}$ |
>     | RMU | $0.5112_{±0.0287}$ | $0.5908_{±0.0255}$ |
>     | NPO | $0.5628_{±0.0893}$ | $0.5041_{±0.0112}$ |
>     | SimNPO | $0.4895_{±0.0512}$ | $0.6519_{±0.0188}$ |
>     | Prompt | $0.3389_{±0.0987}$ | $0.4712_{±0.1356}$ |
>     | GUARDRAIL | $0.3701_{±0.0623}$ | $0.5694_{±0.0765}$ |
>     | ECO | $0.5203_{±0.0341}$ | $0.6635_{±0.0812}$ |
>     | **Ours (PoRT)** | $\textbf{0.8465}_{±0.0156}$ | $\textbf{0.6718}_{±0.0098}$ |
>
>     _Note: For brevity, we only show the main metrics here. In the revised manuscript, we have already updated Tables 7 and 8 in the appendix with the full set of metrics and their statistical uncertainties across all splits._
>
>
> * **Results on WMDP (Zephyr-7b-beta)**
>
>     | Method | Bio Acc.(%) † | Chem Acc.(%)  †  | Cyber Acc.(%)  †| MMLU(%)  $\uparrow$ |
>     | :--- | :---: | :---: | :---: | :---: |
>     | Original | $64.3_{±1.2}$ | $48.5_{±1.5}$ | $43.1_{±1.6}$ | $58.7_{±0.1}$ |
>     | GA | $62.0_{±1.5}$ | $47.1_{±1.4}$ | $42.5_{±1.7}$ | $56.9_{±0.4}$ |
>     | GD | $56.6_{±1.8}$ | $44.6_{±1.9}$ | $36.7_{±2.2}$ | $52.4_{±0.8}$ |
>     | NPO | $62.0_{±1.4}$ | $47.5_{±1.3}$ | $42.6_{±1.8}$ | $57.2_{±0.3}$ |
>     | SimNPO | $46.5_{±2.1}$ | $40.7_{±2.0}$ | $33.9_{±2.5}$ | $49.2_{±1.1}$ |
>     | RMU | $29.5_{±1.2}$ | $47.3_{±1.5}$ | $27.8_{±1.3}$ | $57.2_{±0.2}$ |
>     | Prompt | $63.2_{±1.1}$ | $43.9_{±1.7}$ | $44.2_{±1.5}$ | $57.9_{±0.5}$ |
>     | GUARDRAIL | $51.8_{±1.6}$ | $39.0_{±2.0}$ | $34.7_{±2.3}$ | $57.9_{±0.6}$ |
>     | ECO | $24.7_{±0.5}$ | $26.5_{±0.6}$ | $24.4_{±0.5}$ | $58.6_{±0.2}$ |
>     | **Ours (PoRT)** | $\textbf{25.0}_{±0.4}$ | $\textbf{25.6}_{±0.5}$ | $\textbf{24.8}_{±0.4}$ | $\textbf{58.7}_{±0.1}$ |
>
>     †: For WMDP subsets, best results (closest to 25%) are bolded. $\uparrow$: For MMLU, higher results are bolded.
>
> Accordingly, to maintain the readability of the main text, we have included the full statistical analysis in Appendix D.4.3 & D.5.5 of the revised manuscript. We also reference these statistical uncertainties in the main experimental sections 4.2 and 4.3 to contextualize our findings.
>
> ***

---

> ### Author Response · Authors · 2025-11-20
>
> > **W3 (Minor comment):** The definitions of Forget Probability, ROUGE, and 1-TR for Figure 2 appear only in the appendix (Line 310). A brief, early description would help readers follow the results without jumping ahead.
>
> We thank you for this helpful suggestion and careful reading. In the original submission, space constraints meant we only gave a brief description of these metrics in the Figure 2 caption, which made their meaning somewhat unclear. In the revised manuscript, we have added a clearer explanation in Section 2.2 immediately before Figure 2: `Forget Probability (Prob) measures the model’s confidence in generating the ground-truth answer on the forget set, ROUGE-L Recall (ROUGE) captures the lexical overlap between the generated and ground-truth answers, and Truth Ratio (TR) reflects the model’s ability to distinguish correct from incorrect answers.`
>
> These additions will make Figure 2 self-contained and allow readers to interpret the metrics without needing to consult the appendix.
>
>
> ***
>
> We sincerely appreciate the constructive feedback you have provided and hope that our responses and the modifications to the manuscript adequately address your insightful feedback and increases your impression and confidence in our work. We are happy to provide any further clarifications if needed.
>
> ### References
>
> [1] Maini et al., "TOFU: A Task of Fictitious Unlearning for LLMs," arXiv 2024.
>
> [2] Li et al., "The WMDP Benchmark: Measuring and Reducing Malicious Use with Unlearning," ICML 2024.
>
> [3] Liu et al., "Large Language Model Unlearning via Embedding-Corrupted Prompts," NeurIPS 2024.
>
> [4] Dorna al., "OpenUnlearning: Accelerating LLM Unlearning via Unified Benchmarking of Methods and Metrics," arXiv 2025.

---

> ### Author Response · Authors · 2025-11-25
> **Follow-up on our Rebuttal**
>
> Dear Reviewer cWjy,
>
> Thank you again for your positive assessment and constructive review.
>
> As the discussion phase progresses, we respectfully inquire if our rebuttal has satisfactorily addressed your comments. Under your insightful feedback, we have further strengthened the empirical rigor and clarity of our manuscript.
>
> We are eagerly looking forward to your response and remain fully available if you have any further questions.
>
> Best regards,
> The Authors

---

> > ### Comment · Reviewer_cWjy · 2025-11-26
> >
> > Dear authors,
> >
> > Thank you for your helpful responses. I will keep my overall positive rating.

---

> ### Author Response · Authors · 2025-11-26
>
> Dear Reviewer cWjy,
>
> We are truly grateful for your prompt reply and your continued support of our work. It is encouraging to know that our responses have addressed your concerns.
>
> We deeply value the time and effort you invested in reviewing our paper. Your insightful suggestions have been instrumental in significantly strengthening the empirical rigor of our manuscript. We have poured immense effort into refining this work, and we believe the revised version is now much more robust and comprehensive thanks to your guidance.
>
> If these improvements merit a higher rating in your view, we would be deeply grateful and encouraged.
>
> Thank you once again for your valuable time and constructive engagement.
>
> Best regards,
> The Authors

---

### Official Review · Reviewer_o7ME · 2025-10-31

**Soundness:** 2
**Presentation:** 3
**Contribution:** 2
**Rating:** 4
**Confidence:** 4

**Summary:**

This paper proposes the PoRT (Post judgment and multi-Round Thinking) framework to address the vulnerability of existing LLM unlearning methods under adversarial attacks. The study reveals systemic deficiencies in mainstream pre-filtering approaches: simple prefix attacks can induce up to a 1,150-fold surge in information leakage, while composite question attacks cause accuracy on hazardous knowledge to rebound from the 24.9% random-guess baseline to as high as 67.0%. The PoRT framework comprises three core modules: (1) In-Context Prompt Cleaning (IPC), which leverages LLMs' in-context learning capabilities to dynamically clean adversarial inputs; (2) Post Judgment mechanism, which jointly evaluates cleaned question-answer pairs; and (3) Selective Multi-round Thinking (SMT), which triggers self-correction for low-confidence outputs. Extensive experiments on the TOFU and WMDP benchmarks demonstrate PoRT's superior performance compared to baselines.

**Strengths:**

1. **High-quality writing.** The paper is well-written with clear presentation and logical flow throughout.

2. **Well-motivated framework design.** The approach effectively transforms unlearning from simple rejection to contextual reasoning and continuous refinement, representing a meaningful paradigm shift from pre-filtering to post-judgment evaluation.

3. **Comprehensive experimental validation.** The evaluation is thorough and systematic, spanning 10 representative LLMs, two major benchmarks (TOFU and WMDP), multiple data splits, three types of adversarial attacks, and multi-dimensional evaluation metrics.

**Weaknesses:**

1. **Methodological contradiction.** While PoRT criticizes pre-filtering methods for relying on "superficial input analysis", its own post-judgment classifier is fundamentally a supervised learning model that similarly depends on patterns observed in training data. When confronted with novel attacks outside the training distribution (e.g., potential future multimodal adversarial attacks), the classifier may fail in similar ways to the pre-filtering methods it criticizes.

2. **Limited technical novelty.** All core components represent applications or combinations of existing techniques rather than algorithmic innovations. The IPC module essentially performs data cleaning, the post-judgment classifier is implemented using the existing CCL-SC method, and SMT relies on carefully engineered prompts. The overall contribution appears more engineering-focused than algorithmically innovative, lacking methodological advances beyond systems integration.

3. **Questionable generalization capability.** The post-judgment classifier's ability to generalize remains unclear. All validation is conducted on in-domain data with supervised training and evaluation on identically distributed test sets. The paper provides no evidence of performance on out-of-distribution data, raising concerns about robustness to distribution shifts in real-world deployment scenarios.

**Questions:**

1. **Potential error in Table 4.** PoRT does not achieve the best results in Table 4, yet two entries are bolded as if they were[1]. Specifically, ECO achieves the best results on Bio (24.7% vs. PoRT's 25.1%) and Cyber (24.4% vs. PoRT's 24.8%), but the authors bold their own results. Whether this is an unintentional error or not, the authors benefit from this misrepresentation. **I must bring this discrepancy to the attention of all reviewers and the Area Chair to ensure accurate evaluation.**

2. **Lack of failure case analysis.** The paper would benefit significantly from analyzing failure modes where PoRT's post-judgment classifier and SMT module both fail to prevent information leakage. Understanding the boundaries of the method's effectiveness is crucial for assessing its real-world applicability and guiding future improvements.

---

> ### Author Response · Authors · 2025-11-20
>
> Dear Reviewer o7ME,
>
> We thank you for the thorough review and for acknowledging our **clear writing, well-motivated framework, and extensive experiments**. We will address your insightful questions on our methodology and contributions below.
>
> ***
>
> > **W1: Methodological contradiction.**  &  **W3: Questionable generalization capability.**
>
> We thank the reviewer for raising these insightful points. We address these related concerns together to clarify that **PoRT involves no methodological contradiction; instead, its unique "Methodological Design" fundamentally differs from pre-filtering, ensuring robustness and generalization even on out-of-distribution (OOD) data.**
>
> Our critique of pre-filtering is not that it uses supervision, but that it performs **Binary Classification on Raw Inputs**. This creates a single point of failure where OOD prompts (e.g., noise) easily cross the decision boundary. In contrast, PoRT avoids this "contradiction" through two structural shifts:
>
> *   By evaluating the joint $ (Q, A) $ pair, PoRT anchors its judgment on the semantically stable `Answer`, effectively bypassing the infinite variability of adversarial prompts $ (Q) $ that typically fool input filters.
> *   By employing **Selective Classification** instead of brittle binary predictions, PoRT can **"Abstain"** on OOD inputs rather than forcing a potentially unsafe decision, effectively converting model uncertainty into a safety trigger.
>
> To provide the requested evidence, we evaluated the classifier on distinct OOD attack types not seen during training. Specifically, using a balanced test set ( 500 benign/500 forget per perturbation ) based on the TOFU 10% split, we applied the strict confidence threshold ( $\tau=0.97$ ) from our main experiments. The results below demonstrate **"Robustness via Abstention"**:
>
> | **Perturbation Type** | **Coverage** | **Acc on Covered** | **Abstain Rate** |
> | :--- | :---: | :---: | :---: |
> | None (Baseline) | 84.03% | 98.42% | 15.97% | 83.98% | 98.51% | 16.02% |
> | Rephrased | 79.46% | 98.12% | 20.54% | 83.22% | 98.41% | 16.78% |
> | Synonym-based attack | 76.17% | 97.91% | 23.83% | 82.48% | 98.34% | 17.52% |
> | Keyword fragments | 68.95% | 97.23% | 31.05% | 81.13% | 98.04% | 18.87% |
> | Noise Prefix | 59.82% | 96.11% | 40.18% | 82.26% | 98.21% | 17.74% |
> | Composite Question | 55.27% | 95.79% | 44.73% | 81.86% | 98.13% | 18.14% |
>
> *Note: "Coverage" refers to samples where the classifier predicts compliant/non-compliant; "Abstain Rate" is the percentage of samples sent to SMT for re-thinking.*
>
> These methodological and empirical findings confirm PoRT’s fundamental distinction from pre-filtering. Unlike binary filters that suffer catastrophic failure under distribution shifts, PoRT remains robust on OOD data by correctly identifying uncertainty (via high abstention) while maintaining high accuracy on accepted samples.
>
> Following your valuable comments, we have integrated these results into Appendix E.2 of the revised manuscript to facilitate the review of our classifier's generalization capabilities and robustness.
>
> ***
>
> > **W2: Limited technical novelty.**
>
> We respectfully argue that PoRT goes beyond simple systems integration. Our core contribution is a **fundamental paradigm innovation**—moving from vulnerable "Pre-filtering" to robust "Post-judgment"—**a shift we are honored you recognized as a key strength of our work.** Rather than a simple combination of existing tools, this innovation is powered by **meticulously designed underlying mechanisms absent in existing solutions**, opening a new research direction for robust input-based unlearning. We clarify our methodological contributions from three perspectives:
>
> * Unlike pre-filtering methods that rely solely on the raw query ($Q$), **PoRT evaluates the joint $(Q, A)$ pair.** This reformulation allows the system to detect context-dependent leaks that are structurally invisible to input filters.
>
> * **We adapt sophisticated mechanisms specifically for unlearning robustness, which, **to the best of our knowledge, have not been previously applied in the field of LLM unlearning**:**
>     *   **IPC:** Uses **Syntactic AST Retrieval** (rather than semantic) to resist adversarial paraphrasing.
>     *   **Judgment:** Uses **Selective Classification** to enable the critical "Abstain" signal.
>     *   **SMT:** Mimics **human-like iterative re-thinking** when facing uncertainty, transforming potential leaks into safe responses.
>
> * **These components form a conditional feedback loop rather than a simple combination.** This architecture enables **adaptive robustness**, allowing rapid updates for new threats without retraining the base model.
>
> We sincerely thank the reviewer for this constructive feedback. We believe PoRT represents a pivotal step towards robust, inference-time unlearning, and we hope this paradigm innovation will inspire future research to explore these new architectural directions.
>
>
> ***

---

> ### Author Response · Authors · 2025-11-20
>
> > **Q1: Potential error in Table 4.**
>
> **We must clarify that the bolding in Table 4 is NOT an error, but a correct application of the standard unlearning metric.** For the WMDP benchmark, unlearning effectiveness is measured by **how close the accuracy is to 25%**, which corresponds to random guessing on this 4-choice task. Thus, the “best” score is the one **closest to 25%**, rather than the lowest value.
>
> * **The boldfaced entries in Table 4 follow this criterion (closest to 25%). For example, on Bio, 25.1% (PoRT) is closer to the 25% target than 24.7% (ECO); similarly, on Cyber, 24.8% (PoRT) is closer than 24.4% (ECO).**
>
> **We have mentioned this in line 309 of our original submission**, but we agree that our table layout did not make this evaluation criterion sufficiently explicit. To avoid any ambiguity,  we have revised the caption to explicitly define the evaluation criteria for both metrics. The updated caption reads:`Table 4: Unlearning performance (%) on Zephyr. Bold values indicate best performance: closest to the 25% random baseline for WMDP, and highest accuracy for MMLU…`
>
> Beyond Table 4, we have reinforced the metric definition in the "Experimental Setup" and ensured all WMDP-related tables throughout the manuscript explicitly state that optimal performance is defined by proximity to the 25% baseline.
>
> We appreciate the reviewer’s careful reading and have made this clarification to ensure full transparency.
>
>
> ***
>
> > **Q2: Lack of failure case analysis.**
>
> We thank the reviewer for this insightful suggestion. **In fact, our original submission already included a case study in Appendix F.3, specifically analyzing a scenario where novel attacks cause both the Classifier and SMT to fail, and how extending the IPC module mitigates this.** Inspired by your valuable comment, we have further systematized the "Joint Failure" boundaries into two distinct categories:
>
> * As noted in Appendix F.3, **unseen syntactic forms might bypass IPC normalization.** While this typically leads to a safe increase in abstention, a rare failure occurs if the normalized query appears benign, preventing SMT from triggering. This can be mitigated by extending the IPC demo library and updating the parser.
>
> * **Forgetting novel entities unseen during training presents a second boundary.** The classifier might initially misclassify leaks of these new facts as "compliant," failing to trigger SMT. This challenge is universal to any supervised unlearning approach. However, PoRT's advantage lies in its efficiency: quickly fine-tuning the classifier is far cheaper than re-running costly, parameter-level unlearning on the base model for each new target.
>
> These scenarios represent boundary conditions. We specifically constructed these edge cases to stress-test the system limits. This analysis confirms that even in worst-case scenarios, PoRT's extensible architecture allows for rapid recovery. Accordingly, we have added representative examples to the Appendix F.4 to help assess PoRT's real-world applicability and inspire future improvements.
>
>
> ***
>
> We sincerely appreciate the constructive feedback you have provided and hope that our responses and the modifications to the manuscript adequately address your insightful feedback and increases your impression and confidence in our work. We are happy to provide any further clarifications if needed.

---

> ### Author Response · Authors · 2025-11-24
> **Follow-up on our Rebuttal**
>
> Dear Reviewer o7ME,
>
> Thank you again for your time and constructive feedback.
>
> As the discussion period progresses, we respectfully inquire if our rebuttal has satisfactorily addressed your concerns. In particular, we hope our clarification on Question 1 has resolved the **misunderstanding** regarding the Table 4 metric.
>
> Furthermore, to address your other concerns, we have provided comprehensive empirical evidence and detailed analysis in our response and revised manuscript.
>
> We are eagerly looking forward to your response. Your valuable comments have significantly improved the quality of our manuscript. We remain fully available to engage in further discussion if you have any remaining questions.
>
> Best regards,
> The Authors

---

> > ### Comment · Reviewer_o7ME · 2025-11-26
> >
> > Thank you for the careful response. I apologize for my misunderstanding regarding Table 4. With your clarification, I now understand this error. However, it was merely part of my Question and not a weakness, and it was not a primary factor in my scoring. My main concern remains unaddressed: "Limited technical novelty" (which Reviewer kfwM also raised as a weakness). This results in a lack of contribution to the overall paper. I strongly acknowledge the paper's two existing contributions: "fundamental paradigm innovation" and "extensive experiments." However, as a methodology paper, the specific methods proposed are essentially a combination of existing approaches, lacking innovation. I believe it has not yet reached the standard for publication at a top-tier ML conference like ICLR. I hope that before the next submission, you will either reposition the paper's focus or propose innovative algorithms building upon the current framework. I will keep my score.

---

> > > ### Author Response · Authors · 2025-11-26
> > >
> > > Dear Reviewer o7ME,
> > >
> > > We sincerely appreciate your prompt reply and your acknowledgment of our clarification regarding Table 4. We are glad this specific misunderstanding is resolved.
> > >
> > > Regarding the remaining concern about "Limited technical novelty", we respect your perspective. However, we respectfully find it difficult to reconcile the acknowledgment of "fundamental paradigm innovation" and "extensive experiments" with the conclusion of "lack of contribution." We argue that establishing a new paradigm, guaranteeing robustness, offering a lightweight framework, and enabling rapid, low-cost adaptation to emerging threats collectively constitute a meaningful contribution.
> > >
> > > We submit that PoRT explores a distinct methodological path by shifting to a "post-judgment" paradigm. Furthermore, the underlying mechanisms have not been previously applied in the LLM unlearning domain; they were meticulously designed to address specific SOTA failure modes, representing purposeful engineering rather than a simple combination.
> > >
> > > Ultimately, we hope this work offers a new research direction and inspires future efforts to address critical robustness challenges in LLM unlearning.
> > >
> > > We thank you again for your time and detailed engagement with our paper, which have been very helpful in improving the quality of our work.
> > >
> > > Best regards,
> > > The Authors

---

> > > > ### Comment · Reviewer_o7ME · 2025-11-26
> > > >
> > > > Thanks for the author's further response.
> > > >
> > > > I acknowledge the authors' proposed shift to a "post-judgment" paradigm, which represents a valuable high-level paradigm contribution. However, the framework built upon this paradigm lacks sufficient novelty and appears more as a combination of existing techniques. If the authors could identify problems unique to this paradigm and propose corresponding distinctive solutions, the framework-level design would be more innovative and convincing, rather than presenting a simple composition of existing methods as it currently does.
> > > >
> > > > Additionally, I have reviewed the comments from the other three reviewers, and none of them explicitly highlighted the framework as innovative. As a methodology paper, this work lacks sufficient innovation and contribution at the algorithmic design level, which constitutes a major concern. So, I still decided to keep my score.

---

> ### Author Response · Authors · 2025-11-26
>
> Dear Reviewer o7ME,
>
> We sincerely thank you for your continued engagement and explicit acknowledgment of our **"valuable high-level paradigm contribution."** Regarding your remaining concerns about innovation, we respectfully offer the following clarifications:
>
> > Paradigm-Specific Re-engineering (Not Simple Combination)
>
> We respectfully argue that PoRT is not a simple combination of existing tools. Each component was **meticulously re-engineered** to address the specific constraints of the post-judgment paradigm:
> *   **IPC:** We engineered Syntactic AST Retrieval (vs. semantic) to decouple attack syntax from semantics, creating a novel structural normalization layer.
> *   **Classifier:** We re-engineered CCL-SC by integrating LLM2Vec to encode $(Q, A)$. Using specific "Original" vs. "Retain" contrasts, the confidence-weighted loss effectively distinguishes subtle leakage from hallucinations, making the "Abstain" mechanism mathematically viable.
> *   **SMT:** This is a Cognitive Refinement Loop dynamically triggered by semantic ambiguity to perform reflexive rewriting. This structural innovation directly realizes the "contextual reasoning" strength acknowledged by you and Reviewer cWjy.
>
> These redesigns create a tightly coupled system where the contribution stems from **system-level methodological innovation**.
>
> >Addressing Unique Paradigm Challenges
>
> You correctly pointed out the need to identify unique problems and solutions. PoRT addresses specific challenges inherent to the post-judgment paradigm:
> *   **Unique Problem:** *Post-Generation Uncertainty* (generated answers are neither clearly safe nor unsafe).
>     *   **$\to$** Our **Confidence-Aware Selective Abstention** mechanism.
> *   **Unique Problem:** *Content may contain subtle semantic conflicts that simple rejection cannot fix.*
>     *   **$\to$** **Dynamically triggered by ambiguity to perform "reflexive rewriting"**.
> *   **Unique Problem:** *Adversarial Structural Noise* (attacks bypassing semantic filters).
>     *   **$\to$** **Syntactic Normalization via IPC**.
> *   **Unique Problem:** *Evolving Attack Surfaces.*
>     *   **$\to$** **Inference-time Adaptation** via demo library updates.
>
> Just as paradigms like ICL and CoT [1, 2] were proposed before every specific problem was solved, we believe PoRT establishes a foundational direction. We hope this work inspires future research to further perfect frameworks under this new paradigm.
>
> >Consensus on Innovation
>
> We respectfully correct the statement that *"none of them explicitly highlighted the framework as innovative."* Multiple reviewers specifically identified and validated the novel design elements of PoRT:
> *   **Reviewer kfwM:** Explicitly characterized PoRT as **"a novel unlearning framework,"** recognizing the architectural contribution.
> *   **Reviewer 11Cq:** Praised **"Selective classification with abstain is an interesting safety control; pairing it with targeted re-thinking is efficient."** directly validating our classifier and its unique coupling with the SMT module.
> *   **Your Review:** Acknowledged the shift to **"contextual reasoning and continuous refinement,"** which aligns perfectly with our move from raw-query rejection to **SMT-driven reflective rewriting**.
> *   **Reviewer cWjy:** Highlighted the **"post-judgment mechanism"** that leverages reasoning, validating the innovation of our **system-level framework design**.
>
> >Alignment with ICLR Standards
>
> The ICLR guidelines state: `"Submissions bring value... when they convincingly demonstrate new, relevant, impactful knowledge (incl., empirical, theoretical, for practitioners, etc.)."`
>
> We think PoRT's contributions fall within the definition of impactful methodological knowledge. Furthermore, recent top-tier works [3-8] like Self-Refine (NeurIPS'23), ReAct (ICLR'23), MetaGPT (ICLR'24 Oral), and AFlow (ICLR'25 Oral) are celebrated for paradigm/methodological innovation via architectural orchestration, rather than algorithmic/technical invention. PoRT follows this rigorous, accepted path.
>
> ***
> We deeply value the opportunity to discuss our work with you. Regardless of the final score, we cherish your critical feedback, which has significantly helped us refine the paper’s positioning and rigor. We hope this response provides a more comprehensive view of PoRT’s contributions to the community.
>
> ### References
> [1] Language models are few-shot learners. NeurIPS 2020.
>
> [2] Chain-of-thought prompting elicits reasoning in large language models. NeurIPS 2022.
>
> [3] Self-refine: Iterative refinement with self-feedback. NeurIPS 2023.
>
> [4] React: Synergizing reasoning and acting in language models. ICLR 2023.
>
> [5] Self-discover: Large language models self-compose reasoning structures. NeurIPS 2024.
>
> [6] MetaGPT: Meta programming for a multi-agent collaborative framework. ICLR 2024.
>
> [7] Controllable Safety Alignment: Inference-Time Adaptation to Diverse Safety Requirements. ICLR 2025.
>
> [8] AFlow: Automating Agentic Workflow Generation. ICLR 2025.

---

### Official Review · Reviewer_11Cq · 2025-10-31

**Soundness:** 3
**Presentation:** 3
**Contribution:** 3
**Rating:** 4
**Confidence:** 3

**Summary:**

The paper argues that popular pre-filtering unlearning methods are brittle. Simple prefix or composite-question jailbreaks can reverse the intended forgetting. It proposes PoRT, a three-stage inference-time framework, including In-Context Prompt Cleaning, Post-Judgment, and Selective Multi-round Thinking.

**Strengths:**

1. The paper shows pre-filtering’s vulnerabilities under realistic attacks and provides a valuable perspective.
2. The paper proposes a conceptual shift from input-only filters to answer-aware post-judgment, which aligns with where leakage actually manifests.
3. Selective classification with abstain is an interesting safety control; pairing it with targeted re-thinking is efficient.

**Weaknesses:**

1. The selective classifier seems strong, but robustness hinges on augmentation and demo-library coverage.

2. IPC relies on retrieval of few-shot demos; adversarially crafted inputs may steer cleaning or retrieval. More ablations on $k$, instruction choice, and demo selection failure cases would help.

3. ECO is explained and included, but some recent multi-agent or post-hoc critics are only discussed conceptually. They should be compared in experiments.

4. While improvements are compelling, HFQ is author-introduced; cross-validation by third-party evaluators, or human-rater correlation, would make the claims more robust.

**Questions:**

1. How does performance change when attacks are substantially different from training/augmentation?

2. How is the IPC demo library curated/expanded safely to avoid poisoning? Is there a confidence back-off when retrieval is low-similarity or contradictory?

3. The authors claim modest latency overhead vs. ECO; can you break down cost across IPC, post-judgment, and SMT rounds under real-world harmful-prompt prevalence?

---

> ### Author Response · Authors · 2025-11-20
>
> Dear Reviewer 11cq,
>
> We sincerely thank you for your detailed and insightful review. We are grateful for your recognition of our **core ideas and their conceptual novelty**. Your perceptive questions are central to our work, and we address each of them below.
>
> > **W1:** The selective classifier seems strong, but robustness hinges on augmentation and demo-library coverage.
>
> **We humbly clarify that PoRT’s classifier's robustness does not rely on exhaustively covering attacks via augmentation or the demo library.** The selective classifier is trained with augmentation that supports semantic (rather than attack-specific) generalization (**as detailed in Appendix E.2 of our original submission**). We explain this from two perspectives, supported by empirical evidence:
>
> *   **Joint $(Q, A)$ Evaluation:** While adversarial prompts vary wildly, the leaked information in the `Answer` remains relatively constant and detectable. Therefore, **we incorporate the `Answer` as input for the classifier**, ensuring robustness even against unseen prompt attacks.
> *   **Uncertainty-Aware Abstention:** Specifically, using a balanced test set ( 500 benign/500 forget per perturbation ) based on the TOFU 10% split, we applied the strict confidence threshold ( $\tau=0.97$ ) from our main experiments. Empirically, this ensures robustness on OOD data: the Abstain Rate rises to intercept uncertainty, maintaining high Accuracy on Covered samples as shown below:
>
>     | **Perturbation Type** | **Coverage** | **Acc on Covered** | **Abstain Rate** |
>     | :--- | :---: | :---: | :---: |
>     | None (Baseline) | 84.03% | 98.42% | 15.97% | 83.98% | 98.51% | 16.02% |
>     | Rephrased | 79.46% | 98.12% | 20.54% | 83.22% | 98.41% | 16.78% |
>     | Synonym-based attack | 76.17% | 97.91% | 23.83% | 82.48% | 98.34% | 17.52% |
>     | Keyword fragments | 68.95% | 97.23% | 31.05% | 81.13% | 98.04% | 18.87% |
>     | Noise Prefix | 59.82% | 96.11% | 40.18% | 82.26% | 98.21% | 17.74% |
>     | Composite Question | 55.27% | 95.79% | 44.73% | 81.86% | 98.13% | 18.14% |
>
> Accordingly, we have integrated this detailed analysis regarding perturbation types and the corresponding empirical results into Appendix E.2 of the revised manuscript.
>
> ***
>
>
> > **W2:** IPC relies on retrieval of few-shot demos; adversarially crafted inputs may steer cleaning or retrieval. More ablations on *k*, instruction choice, and demo selection failure cases would help.
>
> We thank the reviewer for the attention and suggestions regarding the IPC module. **We respectfully clarify that the IPC module is robust against adversarial inputs, and our original submission already included some relevant experiments and failure case studies.** We address your specific points below:
>
>
> * **IPC is designed to be inherently robust against adversarial steering through structural rather than semantic analysis.** IPC relies on AST-based retrieval (Appendix E.1), which is less sensitive to adversarial paraphrasing compared to semantic-based retrieval.
>
>
> * Comprehensive ablations were included in the original submission, and we add further analyses here for clarity:
>
>     *   **Sensitivity to $k$:** Figure 9(b) has already provided a sensitivity analysis, supporting our choice of *k*=3.
>     *   **For instruction choice:** Table 14 (Appendix E.1) shows that an instruction is required for stable cleaning. Following the same setup, we examined several instruction styles (on Llama-2-13b-chat-hf):
>
>         | **Instruction Variant** | **Example** | **Similarity (↑)** |
>         | :--- | :--- | :--- |
>         | **Ours (Original)** | "You are a prompt cleaner. Given a noisy query, output ONLY the cleaned question." | **0.9961** |
>         | Variant A (Polite, verbose) | "Please refer to the following examples. For the last input Query, generate a standard Processed Prompt..." | 0.9750 |
>         | Variant B (Direct) | "Output ONLY the cleaned question...." | 0.9395 |
>
> *   **"Demo selection failure" is effectively mitigated by our framework's extensibility (detailed in original Appendix F.3).** Failures typically stem from novel structures, causing graceful degradation (see **Q1**). Full performance is rapidly restored by updating the demo library and AST parser, while Post-Judgment and SMT modules serve as immediate safeguards even prior to such updates.
>
> Accordingly, as the ablation studies on $k$ and demo selection failure cases were already included in the original submission, we have specifically added the instruction choice results to Appendix E.1 of the revised manuscript.
>
> ***

---

> ### Author Response · Authors · 2025-11-20
>
> > **W3:** ECO is explained and included, but some recent multi-agent or post-hoc critics are only discussed conceptually. They should be compared in experiments.
>
>
> We thank the reviewer for highlighting recent works. **We have newly conducted a direct experimental comparison with ALU [1] below. In contrast, we clarify that "post-hoc critics" fall under the category of input-based methods, which were already extensively evaluated in our original submission.**
>
>
> * We initially excluded ALU from the original submission as the unavailability of its codebase precluded a fair reproduction. To address your comment, we perform a direct comparison here using their reported results on the 5 overlapping models within the WMDP benchmark:
>
>     | **Model** | **Method** | **Bio†** | **Chem†** | **Cyber†** | **MMLU (↑)** |
>     | :--- | :--- | :--- | :--- | :--- | :--- |
>     | **deepseek-moe-16b**| ALU / PoRT (Ours) | 25.8% / **25.2%** | 26.1% / **24.5%** | 25.5% / **25.2%** | N/A  / **48.0%** |
>     | **Llama-2-7b-hf** | ALU / PoRT (Ours) | 24.2% / **25.4%** | **26.8%** / 23.0% | 24.8% / **24.9%** | N/A / **46.5%** |
>     | **Llama-2-13b-hf** | ALU / PoRT (Ours) | 26.5% / **25.0%** | **24.5%** / 23.0% | 24.6% / **24.9%** | N/A / **53.0%** |
>     | **Llama-3-8b**| ALU / PoRT (Ours) | **24.6%** / 26.8% | 24.1% / **24.8%** | **25.0%** / 26.5% | 57.8% / **64.9%** |
>     | **Qwen1.5-7B** | ALU / PoRT (Ours) | 25.9% / **24.8%** | 27.8% / **23.0%** | 25.6% / **24.8%** | N/A  / **37.6%** |
>     | **Qwen1.5-14B** | ALU / PoRT (Ours) | **24.9%** / 24.8% | **24.8%** / 26.5% | **25.1%** / **25.1%** | N/A / **65.8%** |
>
>
>     †: For WMDP subsets, best results (closest to 25%) are bolded. N/A: Not Available in the ALU paper.
>
>     The results demonstrate that **PoRT outperforms ALU across three critical dimensions:**
>
>     * PoRT achieves **superior unlearning**, matching or exceeding ALU's proximity to the random baseline in 11 out of 18 comparisons.
>     * PoRT **preserves significantly more general knowledge**, outperforming ALU by a wide margin (64.9% vs. 57.8%) on the only reported MMLU benchmark.
>     * PoRT is substantially **more efficient** compared to ALU's resource-intensive multi-agent pipeline.
>
> * Regarding "post-hoc critics," the ALU paper categorizes inference-time approaches as "post-hoc methods." This corresponds exactly to the class of input-based methods (e.g., ECO) that we have already extensively analyzed in our work.
>
> Following your valuable suggestion, we have integrated this comparative analysis into Appendix D.5.4 of the revised manuscript.
> ***
>
> > **W4:** While improvements are compelling, HFQ is author-introduced; cross-validation by third-party evaluators, or human-rater correlation, would make the claims more robust.
>
>
> We thank the reviewer for raising this point. **We demonstrate that holistic scores from an independent third-party evaluator show the same trend as HFQ.** While full human evaluation is beyond the scope of this submission, we follow recent practice by using GPT-5.1 as an independent third-party evaluator to provide the robust validation requested:
>
> | **Method** | **Behavior** | **LLM Judge Score** | **HFQ** | **LLM Judge's Qualitative Assessment** |
> | :--- | :--- | :--- | :--- | :--- |
> | Original | Data Leakage | 5.34 | 0 | "Fails completely by directly leaking the forgotten fact." |
> | GA | Catastrophic Forgetting | 12.52 | 9.47 | "Avoids leakage but at the cost of generating irrelevant and unhelpful content." |
> | ECO | Nonsensical Output | 18.16 | 17.74 | "Produces incoherent and nonsensical text; a form of system failure." |
> | NPO | Partial Unlearning | 58.76 | 56.59 | "A reasonable attempt but often still hints at the information or is overly evasive." |
> | **PoRT (Ours)** | Plausible Unlearning| 91.53 | 90.16 | "Successfully removes the fact while providing a coherent and plausible alternative response." |
> | Retain | Ideal Unlearning | 93.82 | 100 | "Generates a high-quality, perfectly unlearned response." |
>
> This consistency validates HFQ as **a meaningful complement to standard metrics** (Table 7), where PoRT also excels. Given that standard leakage metrics capture limited semantic nuances [2, 3], HFQ offers a necessary perspective. We have integrated this analysis into Appendix F.2 and discussed human validation in the Appendix G "Future Work".
>
> ***

---

> ### Author Response · Authors · 2025-11-20
>
> > **Q1:** How does performance change when attacks are substantially different from training/augmentation?
>
> We thank the reviewer for raising this important question. **The results demonstrate that unlike baselines which suffer catastrophic failure under unseen attacks, PoRT avoids such collapse and adapts quickly to restore full performance.**
>
> While our response to **W1** demonstrated how the classifier's performance changes when facing unseen data, here we focus on the performance evolution of the **entire PoRT framework** (measured by HFQ).
>
> *  **Immediate performance degrades gracefully rather than catastrophically.** Under a novel jailbreak attack, HFQ dropped only modestly **(0.8474 $\to$ 0.7520)**. This resilience holds because joint $(Q, A)$ evaluation and uncertainty-aware abstention still trigger SMT to refine responses, ensuring safety even on unseen data.
> *   **Full performance can be rapidly restored via low-cost adaptation.** As shown in Appendix F.3, PoRT adapts to new threats via lightweight updates to auxiliary components (demo library/parser) rather than costly retraining, ensuring scalable robustness:
>
>     | **Condition** | **Method** | **HFQ (↑)** |
>     | :--- | :--- | :--- |
>     | Standard (No Attack) | PoRT (Original) | 0.8474 |
>     | Under Jailbreak Attack | PoRT (Original) | 0.7520 |
>     | Adapted to Jailbreak Attack| PoRT (Adapted) | 0.8355 |
>
> Following your valuable comments, we have added this case study on "Jailbreak Attacks" and the adaptability analysis to Appendix F.3 of the revised manuscript.
> ***
>
>
> > **Q2:** How is the IPC demo library curated/expanded safely to avoid poisoning? Is there a confidence back-off when retrieval is low-similarity or contradictory?
>
> **We sincerely thank the reviewer for this professional and forward-looking view.** Currently, the IPC library is maintained via **offline manual curation** to strictly prevent poisoning, and we rely on the downstream **Post-Judgment Classifier** to handle low-similarity retrieval (triggering "Abstain" if generation quality suffers) rather than an explicit retrieval back-off.
>
> We respectfully submit that these specific challenges arise precisely because PoRT introduces a **new paradigm**—shifting from simple "rejection" to "cognitive processing." While pre-filtering methods avoid these complexities by being overly simplistic, our post-judgment framework introduces new research dimensions for robustness. We fully agree that automating safe expansion and implementing retrieval-level back-off are essential next steps to mature this paradigm, and we hope our work inspires the community to explore these new directions.
>
> Inspired by your valuable feedback, **we have explicitly added "Automated Safe Curation" and "Retrieval Confidence Back-off" as key directions in the "Limitations and Future Work" section of the revised manuscript**.
>
>
> ***
>
>
> > **Q3:** The authors claim modest latency overhead vs. ECO; can you break down cost across IPC, post-judgment, and SMT rounds under real-world harmful-prompt prevalence?
>
> Thank you for your question on PoRT's practical efficiency. The table below provides a detailed latency breakdown of PoRT's components versus ECO's baseline. To simulate the "real-world harmful-prompt prevalence" you mentioned, we created additional test splits from TOFU with harmful prompt rates down to 0.1%. All latencies, measured in milliseconds (ms), were benchmarked with Llama-2-7b.
>
>
> | Harmful Rate | ECO Latency | IPC | Post-Judgment | SMT | Total PoRT Latency | Overhead |
> | :--- | :--- | :--- | :--- | :--- | :--- | :--- |
> | **10%** | 370.70 | 372.72 | 3.43 | 18.87 | 395.02 | 6.56% |
> | **5%** | 370.51 | 371.63 | 3.35 | 9.45 | 384.43 | 3.76% |
> | **1%** | 371.08 | 372.28 | 3.32 | 1.89 | 377.49 | 1.73% |
> | **0.1%** | 370.85 | 370.75 | 3.28 | 0.19 | 374.22 | 0.91% |
>
>  Our reported 6.56% overhead represents a conservative upper bound measured in a worst-case scenario. In more realistic settings, which we argue are represented by the **1% and 0.1% prevalence** splits, the variable SMT cost diminishes significantly, making PoRT's latency nearly identical to single-pass methods like ECO. We have added these detailed results to the Appendix D.6.
>
>
> ***
>
> We sincerely appreciate the constructive feedback you have provided and hope that our responses and the modifications to the manuscript adequately address your insightful feedback and increases your impression and confidence in our work. We are happy to provide any further clarifications if needed.
>
>
> ### References
> [1] Sanyal & Mandal, "Agents Are All You Need for LLM Unlearning," COLM 2025.
>
> [2] Mekala et al., "Alternate Preference Optimization for Unlearning Factual Knowledge in Large Language Models," COLING 2025.
>
> [3] Yuan et al., "A Closer Look at Machine Unlearning for Large Language Models," ICLR 2025.

---

> ### Author Response · Authors · 2025-11-24
> **Follow-up on our Rebuttal**
>
> Dear Reviewer 11Cq,
>
> Thank you again for your encouraging review. We deeply appreciate your positive assessment of our work's soundness, presentation, and contribution, particularly your recognition of the conceptual novelty and paradigm shift PoRT introduces.
>
> As the discussion phase progresses, we respectfully inquire if our response has satisfactorily addressed your concerns. In particular, we hope the newly added comparison with ALU (W3), the third-party validation for HFQ (W4), and the detailed OOD robustness analysis (W1) have further enhanced your positive impression of our work.
>
> **We are eagerly looking forward to your response.** We greatly value your constructive suggestions and remain fully available to engage in further discussion if you have any remaining questions.
>
> Best regards,
> The Authors

---

> > ### Comment · Reviewer_11Cq · 2025-11-26
> >
> > I appreciate the detailed response. It addressed most of my concerns, so I would like to increase my rating.

---

> > > ### Author Response · Authors · 2025-11-26
> > >
> > > Dear Reviewer 11Cq,
> > >
> > > Thank you very much for your follow-up and for taking the time to reconsider your evaluation. We are glad that our responses helped clarify your concerns.
> > >
> > > We greatly appreciate your decision to raise the score for our submission and your thoughtful comments, which have been very helpful in improving the quality of our work.
> > >
> > > Kind regards,
> > > The authors

---

### Official Review · Reviewer_kfwM · 2025-11-01

**Soundness:** 3
**Presentation:** 3
**Contribution:** 3
**Rating:** 6
**Confidence:** 2

**Summary:**

The paper introduces PoRT, a novel unlearning framework for large language models that enhances compliance and safety by removing sensitive knowledge without harming overall performance. By combining data cleaning, post-judgment evaluation, and selective multi-round thinking, PoRT significantly improves robustness against adversarial attacks and maintains strong unlearning effectiveness.

**Strengths:**

The writing flow and structure is easy to follow. Comparisons across pre-filtering methods under adversarial attacks support a clearer motivation.

Comprehensive experiments are conducted to validate PoRT's effectiveness and efficiency, providing valuable insight in LLM Unlearning without params altering.

**Weaknesses:**

The techniques and methods incorporated into the framework are integrated in a rather straightforward manner, making the overall contribution appear somewhat trivial.

**Questions:**

null

---

> ### Author Response · Authors · 2025-11-20
>
> Dear Reviewer kfwM,
>
> We thank you for your insightful and constructive feedback. We appreciate your positive assessment of our work's **motivation, clear structure, and extensive evaluation**. Below, we address your concerns to further clarify our contributions.
>
> > **W1:** The techniques and methods incorporated into the framework are integrated in a rather straightforward manner, making the overall contribution appear somewhat trivial.
>
> We thank the reviewer for the valuable comments. We respectfully argue that our core contribution is a **fundamental paradigm innovation** from vulnerable "Pre-filtering" to robust "Post-judgment," an innovation recognized as a key strength by all three other reviewers. We submit that the workflow's high-level simplicity conceals **sophisticated, non-trivial mechanisms designed to address critical failure modes**, thereby exploring a distinct and promising research avenue. We humbly respond to your concerns from the following perspectives:
>
> * **Our core contribution is a paradigm innovation from pre-filtering to a robust post-judgment model.** Unlike pre-filtering methods that only judge ambiguous raw queries `(Q)`, PoRT assesses the high-information joint `(Q, A)` pair, where leakage is most evident. This shift in what is being evaluated enables far more accurate judgments.
>
> * **PoRT's design is non-trivial, both within and between its components.**
>
>     * **Within components, our design is **sophisticated** and engineered for robustness, not the most straightforward options.** For example, IPC's demonstration retrieval uses syntactic structure matching, not simple semantic similarity, to resist adversarial paraphrasing. Similarly, our classifier is selective, not a binary model, enabling it to abstain on uncertain inputs—a crucial feature for safety.
>
>     * **Between components, the modules form a synergistic pipeline, not a simple stack.** Each stage critically enables the next in a non-straightforward manner: IPC provides a clean signal for the classifier, whose "abstain" verdict then precisely triggers SMT's corrective reasoning. This interdependence is the source of PoRT's holistic robustness.
>
> [Continued below]

---

> ### Author Response · Authors · 2025-11-20
>
> * Below, we present a feature-based comparison that starkly illustrates the architectural innovations of PoRT, which are absent in prior works.
>
>     | Feature Dimension         | Prompt [1] | GUARDRAIL [2] | ICUL [3] | ERASE [4] | ECO [5] | **Ours (PoRT)** |
>     | :------------------------ | :---------------: | :--------: | :---: | :----: | :--: | :-------------: |
>     | **_A. Core Mechanism_**    |                   |            |       |        |      |                 |
>     | Intervention Stage        |   In-inference    | Pre-inference | In-inference | In-inference | Pre-inference | **Post-inference** |
>     | Primary Method            | Instruction Injection | Input Filtering | Contextual Examples | Example Selection | Embedding Perturbation | **Clean, Judge, Correct** |
>     | **_B. Response Quality_**  |                   |            |       |        |      |                 |
>     | Response Type             | Uncontrolled    | Template Refusal | New Content | New Content | Nonsensical Content | **High-Quality Fiction** |
>     | Response Naturalness      |         ❌         |      ❌       |   △   |   △    |  ❌   |     **✅**      |
>     | **_C. System Design_**     |                   |            |       |        |      |                 |
>     | Judgment Module           |  Implicit (LLM)   | Explicit (External) | Implicit (LLM) | Implicit (Clustering) | Explicit (External) | **Explicit (External)** |
>     | Judgment Basis            |      Query (Q)      |   Query (Q)  | Query (Q) | Query (Q) | Query (Q) | **Joint (Q, A) Pair** |
>     | Iterative Self-Correction |         ❌         |      ❌       |   ❌   |   ❌    |  ❌   |     **✅**      |
>     | Selective Trigger         |         ❌         |      ❌       |   ❌   |   ❌    |  ❌   |     **✅**      |
>     | Dynamic Adaptability      |         △         |      ❌       |   △   |   ❌    |  ❌   |     **✅**      |
>
>     **Legend:**  ✅: Fully supported; △: Partially or conditionally supported; ❌: Not supported
>
> In summary, PoRT transcends dominant pre-filtering methods to **explore a distinct and promising avenue for robust unlearning**. We hope this post-judgment paradigm inspires future work, and have accordingly detailed this architectural discussion in a new appendix section on **"Methodological Contributions and Future Work"**.
>
> ***
>
> We sincerely appreciate the constructive feedback you have provided and hope that our responses adequately address your insightful feedback and increases your impression and confidence in our work. We are happy to provide any further clarifications if needed.
>
> ### References
> [1] Lynch et al., "Eight Methods to Evaluate Robust Unlearning in LLMs," arXiv 2024.
>
> [2] Thaker et al., "Guardrail Baselines for Unlearning in LLMs," arXiv 2024.
>
> [3] Pawelczyk et al., "In-Context Unlearning: Language Models as Few-Shot Unlearners," ICML 2024.
>
> [4] Muresanu et al., "Fast Exact Unlearning for In-Context Learning Data for LLMs," ICML 2025.
>
> [5] Liu et al., "Large Language Model Unlearning via Embedding-Corrupted Prompts," NeurIPS 2024.

---

> ### Author Response · Authors · 2025-11-25
> **Follow-up on our Rebuttal**
>
> Dear Reviewer kfwM,
>
> We are writing to respectfully follow up on our previous response.
>
> As the discussion phase progresses, we respectfully inquire if our clarification of the work's contributions is sufficient to address your main concern regarding technical novelty.
>
> We sincerely value your positive rating and are happy to provide any further clarifications if needed.
>
> Best regards,
> The Authors

---

> ### Author Response · Authors · 2025-11-27
>
> Dear Reviewer kfwM,
>
> As the discussion phase progresses, we wanted to touch base regarding your concern about the contribution of our work.
>
> As detailed in our response and recent exchange with Reviewer o7ME, PoRT is a meticulous design optimized both within and between its components, rather than a simple combination. Techniques like Syntactic AST Retrieval and Selective Classification were specifically re-engineered for this task and are, to our knowledge, novel applications in the unlearning domain. Moreover, we emphasize the **fundamental paradigm innovation** of shifting from pre-filtering to post-judgment. This architectural contribution is recognized by the other three reviewers and aligns with the standards of recent top-tier works.
>
> We are eagerly looking forward to your response. If these points help solidify your positive assessment or potentially merit a higher rating, we would be deeply grateful and encouraged.
>
> Best regards,
> The Authors

---

### Author Response · Authors · 2025-11-29
**[Summary for AC 2/2 - Part B] Comprehensive Report**

(Continued from [Summary for AC 2/2 - Part A])
*   **Reviewer o7ME (4 $\to$ 4)**
    *   **Initial Assessment:** Strongly acknowledged **"valuable high-level paradigm contribution, well-motivated framework design, and extensive experiments"**. **Concerns:** **Misunderstood** Table 4 as "misrepresentation," worried about OOD robustness and overall contribution, requested case studies.
        *   **$\to$ Nov 20 (`Initial Response`):** Clarified Table 4 metric definitions; pointed out case studies were already included; added robustness analysis; argued for the paper's contribution.
        *   **$\to$ Nov 26 (`Critical Turning Point`):** Reviewer **explicitly retracted the allegation**: ***"I apologize for my misunderstanding regarding Table 4... I now understand this error."*** While maintaining the score due to contribution concerns, they reaffirmed recognition of the **"high-level paradigm contribution."**
        *   **$\to$ Nov 26 (`Further Discussion`):** We systematically refuted the remaining concern from three aspects: **value of paradigm innovation, innovative adaptation of technologies, and ICLR standards/excellent papers**. We pointed out the **logical contradiction in acknowledging "valuable paradigm contribution" and "well-motivated framework" while denying contribution, strongly defending the substantial contribution of our work**.
    *   Of the 5 concerns raised by Reviewer o7ME, only the question regarding overall contribution remains under active discussion, **where we have provided sufficient arguments demonstrating the substantial contribution of our work**. All other issues have been resolved (**especially the critical data misunderstanding**).

*   **Reviewer cWjy (6 $\to$ 6)**
    *   **Initial Assessment:** **Affirmed "post-judgment mechanism, extensive empirical evaluation and well-structured writing"**. **Concerns:** Suggested adding statistical uncertainty and method comparison results.
        *   **$\to$ Nov 20 (`Initial Response`):** Updated all key tables to report **Mean ± Std (5 seeds)**; clarified that comparisons with Model-based methods **were already fully discussed in the original submission**, and further highlighted them in the main text.
        *   **$\to$ Nov 26 (`Confirmation`):** Reviewer responded: *"Thank you for your helpful responses. **I will keep my overall positive rating**."*
    *   **All 3 concerns raised by Reviewer cWjy were resolved, leading to the confirmation of a positive rating.**

**We summarize the improvements made to the paper during the rebuttal period** for your quick review:

*   **We clarified that several analyses requested by reviewers were actually included in our original submission** (Appendices D.4, D.5, F.3, E.2)—including **Model-based comparisons** (cWjy), **Failure Case Studies** (o7ME), and **Generalization Basis** (11Cq). In the revised version, we elevated these results to the main text or highlighted them to ensure visibility.
*   Furthermore, based on constructive feedback, **we made significant improvements to enhance paper quality (all relevant content has been added to the revised manuscript):**
    1.  **New Experiments & Comparisons:**
        *   Key results in **Sec 4.2 & 4.3** and appendix tables now include **Mean ± Std (5 seeds)**.
        *   Added direct comparison with multi-agent unlearning framework (ALU) in **App. D.5.4**, demonstrating PoRT's efficiency and utility advantages.
        *   Added detailed latency breakdown in **App. D.6**, proving minimal deployment overhead.
        *   Added sensitivity analysis for the classifier under OOD attacks in **App. E.2**, proving the effectiveness of the "abstention mechanism."
        *   Added GPT-5.1 based LLM-as-a-Judge evaluation in **App. F.2** to validate HFQ reliability.
    2.  **Clarity & Depth:**
        *   Defined evaluation metrics upfront in **Sec 2.2** to eliminate ambiguity.
        *   Added **Appendix G**, detailing how PoRT's architectural orchestration constitutes system-level innovation.
        *   Systematized analysis of "Joint Failure" boundaries and recovery mechanisms in **App. F.4**.

We hope this detailed record helps you quickly understand the results achieved through our joint efforts with the reviewers, serving as a quick guide for checking relevant details. Thank you again for your review and consideration under this high workload. **We remain fully available for further discussion.**

Best regards,
The Authors

---

### Author Response · Authors · 2025-11-29
**[Summary for AC 2/2 - Part A] Comprehensive Report**

Dear Area Chair,

We sincerely appreciate your willingness to take on this additional workload and thank you for your dedication to the review process. We submit this detailed report to assist you in quickly and comprehensively assessing our work, serving as a quick guide for reviewing relevant details.

This paper proposes PoRT, a novel inference-time framework for LLM Unlearning. By shifting from a vulnerable "Pre-filtering" paradigm to a robust "Post-judgment" paradigm—incorporating Syntactic AST Retrieval, Selective Classification, and Reflective Correction—PoRT addresses the catastrophic failure of SOTA methods under adversarial attacks, achieving enhanced robustness. **Through paradigm innovation and technological adaptation innovation, this work explores a new research direction for LLM Unlearning.**

*   During the discussion period, our work gained further recognition, raising the effective average score from **5.0 to 5.5**.
*   Reviewers identified "Paradigm Innovation," "Extensive Experiments," "Clear Writing," "Valuable Perspective," and "Effective Architecture Design" as key strengths.
*   Issues initially raised regarding "Classifier Robustness," "Supplementary Experiments," "Detailed Case Studies," "Data Misunderstandings," and "Statistical Uncertainty" have been confirmed as resolved during the discussion.

*   The sole remaining divergence lies in Reviewer o7ME's judgment of the paper's overall contribution. We have cited ICLR guidelines and 2024/2025 Oral papers to demonstrate that **paradigm innovation and technological adaptation are widely recognized valuable contributions.** Given that **PoRT's paradigm innovation has been widely acknowledged by reviewers, and we have repeatedly explained the technical adaptations and innovations in detail**, we invite the Area Chair to review our latest response to Reviewer o7ME regarding this point. **We remain fully available for further discussion with you.**

**The following record details the evolution of each reviewer's stance and key resolutions, spanning from the initial review stage to the discussion period** (all times in UTC). We solemnly declare that all interactions **strictly adhered to ICLR's double-blind policy** and occurred prior to the public disclosure of the data leak. **Overall, the four reviewers raised a total of 16 concerns; 14 have been fully resolved, leading to positive evaluations and score increases prior to the discussion deadline. Of the remaining two concerns, one is pending feedback, and one is under active discussion:**

*   **Reviewer kfwM (6 $\to$ 6)**
    *   **Initial Assessment:** Highly recognized PoRT as **"a novel unlearning framework"**; praised the **"clear motivation and comprehensive experiments"**. **Sole concern** was that the technical integration seemed straightforward.
        *   **$\to$ Nov 20 (`Initial Response`):** We clarified that PoRT involves **"meticulous re-engineering"** of components rather than simple combination, supported by a new feature comparison table.
        *   **$\to$ Nov 25/27 (`Further Substantiation`):** Further referenced ICLR standards and related papers to reinforce the **contribution claim**.
    *   Reviewer kfwM raised only 1 concern. We have provided a detailed response proving innovation in our technical combination and substantial overall contribution, though we did not receive a reply within the discussion period.

*   **Reviewer 11Cq (4 $\to$ 6)**
    *   **Initial Assessment:** **Affirmed the "conceptual shift, framework design, and the valuable perspective"**. **Concerns:** OOD robustness, IPC anti-interference, and lack of some auxiliary experiments.
        *   **$\to$ Nov 20 (`Initial Response`):** **Clarified that some experiments were already included**, added OOD robustness analysis (**proving the effectiveness of the abstention mechanism**), ALU comparison, and third-party validation.
        *   **$\to$ Nov 26 (`Score Raised`):** Reviewer responded: *"The detailed response... addressed most of my concerns, so **I would like to raise my rating**,"* **raising the score from 4 to 6**.
    *   **All 7 concerns raised by Reviewer 11Cq were resolved, leading to a score increase.**

(Due to character limits, the detailed report continues in [Summary for AC 2/2 - Part B])

---

### Author Response · Authors · 2025-11-29
**[Summary for AC 1/2] Executive Summary**

Dear Area Chair,

We sincerely appreciate your willingness to take on this additional workload and thank you for your dedication. To assist your evaluation, we submit this brief, objective summary of our rebuttal interactions and preliminary results achieved during the discussion period.

**We guarantee strict adherence to ICLR's double-blind policy.** During the discussion period prior to the data leak:

*   Through active discussion, our work gained further recognition, raising the average score from **5.0 to 5.5 (Nov 26, 02:50 UTC)**.
*   All positive resolutions and score increases occurred on **Nov 26 (UTC)**. This is approximately **24 hours prior** to the public disclosure of the data leak (approx. Nov 27 15:00 UTC).
*   We received positive feedback from some reviewers while still actively debating others, evidencing a natural, non-collusive review process.

These interactions reflect scientific rigor; the score increases were **solely the result of our substantial rebuttal efforts and the reviewers' careful deliberation.** While the rollback nullifies the recorded score increase, **we fully respect ICLR's policy adjustment.** We provide this concise summary to outline key pre-deadline developments for your quick review (*a detailed discussion log follows in the next official comment*).

| Reviewer | Score | Summary of Review & Discussion |
| :--- | :---: | :--- |
| **cWjy** | **6 $\to$ 6** | Fully affirmed our **"post-judgment mechanism and extensive empirical evaluation"**. Added statistical uncertainty and comparisons **resolved concerns**; reviewer confirmed to **"keep overall positive rating."** |
| **kfwM** | **6 $\to$ 6** | Highly praised the **"novel unlearning framework, clear motivation, and comprehensive experiments"**. Regarding "straightforward integration," we strongly **demonstrated technical innovation** via component re-engineering, synergy, and feature comparison. However, no further response was received from the reviewer prior to the discussion deadline. |
| **11Cq** | **4 $\to$ 6  (**Nov 26, 02:50 UTC**)**  | Highly recognized our **"valuable perspective, conceptual shift, and efficient framework"**. Added OOD robustness analysis, metric validation, and supplementary experiments; together with clarifications of the original manuscript, these **"addressed concerns"**. **Reviewer raised score to 6.** |
| **o7ME** | **4 $\to$ 4** | Praised **"valuable high-level paradigm contribution, well-motivated framework design and extensive experiments"**. Following clarification, **retracted the allegation of misrepresentation in Table 4**. Addressing the remaining "insufficient innovation and contribution" concern, we systematically refuted this by citing **paradigm value, technical adaptation, and ICLR standards/Oral papers**, highlighting the **logical inconsistency** between acknowledging our "valuable paradigm contribution" and "well-motivated framework" while simultaneously denying the work's contribution, thus **strongly defending the substantial contribution of our work**. |

We fully understand and respect ICLR's policy adjustments in light of this incident. Thank you again for your consideration under this high workload. We hope this summary provides useful context for your recommendation. **For a comprehensive report covering our paper's core value, detailed reviewer consensus, and the full rebuttal timeline, please refer to our accompanying comment: "[Summary for AC 2/2] Comprehensive Report".**

Best regards,
The Authors

---

### Meta-Review · Area_Chair_NGGa · 2026-01-02

**Summary:**

Unlike pre-filtering methods that rely solely on the query (Q), this paper proposes a new unlearning method which evaluates the joint query-answer pair (Q,A). This reformulation allows the system to detect context-dependent leaks that are structurally invisible to input filters, and thus more robust.

**Reviewer Concerns:**

1. Limited technical novelty. Core components of the method are either applications or combinations of existing techniques.

**Reviewer Scores:**

kfwM would keep the score
11Cq would increase the score
o7ME would increase the score, but not much
cWjy would keep the score

---

### Decision · Program_Chairs · 2026-01-26

Accept (Poster)